# Acute Shear Stress Induces TWIST-Mediated EndMT in Venous Endothelial Cells and Human Long Saphenous Veins

**DOI:** 10.3390/cells14171369

**Published:** 2025-09-02

**Authors:** Shameem S. Ladak, Liam W. McQueen, Kristina Tomkova, Georgia R. Layton, Gavin J. Murphy, Mustafa Zakkar

**Affiliations:** Department of Cardiovascular Sciences, University of Leicester, Leicester LE3 9QP, UKlwm3@leicester.ac.uk (L.W.M.);

**Keywords:** saphenous vein, CABG, EndMT, Dexamethasone

## Abstract

The use of vein grafts in coronary artery bypass graft surgery is complicated by a high late restenosis rate resulting from the development of intimal hyperplasia. The role of changes in haemodynamics on TGFβ-driven endothelial-to-mesenchymal transition (EndMT) is not fully understood. We identified that arterial shear stress can induce TGFβ/SMAD-dependent EndMT in human umbilical vein endothelial cells (HUVECs), which was regulated by TWIST transcription factors (TWIST1&2), as the selective inhibition of TWIST1 or TWIST2 using siRNA-suppressed EndMT. We noted that brief pretreatment of HUVECs with Dexamethasone can modulate EndMT in response to shear stress. Using spatial cell sequencing in human long saphenous vein segments exposed to acute arterial flow, we identified a cluster of cells with both endothelial and smooth muscle-cell (SMC)-like phenotypes, in which TWIST2 was significantly upregulated. We validated the untargeted spatial findings in segments of veins under acute arterial flow ex vivo. We observed that Dexamethasone can suppress EndMT changes in vein segments by suppressing TGFβ/SMAD/TWIST1&2. This suggests that Dexamethasone brief pretreatment can suppress EndMT changes triggered by acute exposure of long saphenous vein segments to arterial haemodynamics by modulating the TGFβ/SMAD/TWIST1&2 pathway.

## 1. Introduction

Coronary artery bypass grafting (CABG) is the standard of care for severe coronary artery disease, with the long saphenous vein (LSV) being the most frequently used conduit [1,2]. However, the long-term success of LSV grafts is limited by the development of intimal hyperplasia (IH), which leads to graft failure and major adverse cardiovascular events [1,3]. It is accepted that IH is a multifactorial process that can be triggered as soon as the vein is harvested. However, the exact mechanisms regulating the pathophysiological changes are still not fully understood. That said, the exposure of LSVs to acute haemodynamics when veins are implanted into arterial circulation can play a pivotal role in such a process [4,5]. Endothelial-to-mesenchymal transition (EndMT) is the process by which endothelial cells (ECs) lose their cell-specific markers and morphology and acquire a mesenchymal cell-like phenotype with a complex orchestration of several signalling pathways involved in regulating this process [6]. EndMT has been implicated in regulating cell phenotype adaptation during vein graft remodelling after grafting into the arterial circulation, with the TGFβ/SMAD signalling pathway playing a pivotal role in regulating this process [7,8]. We previously demonstrated that acute exposure of venous, but not arterial, ECs to arterial shear stress is associated with activating different proinflammatory pathways, including MAPK and NF-kB, resulting in the upregulation of different proinflammatory molecules such as VCAM and MCP-1 [9,10]. Furthermore, arterial but not venous shear activated such responses in venous ECs [11].

Glucocorticoids regulate a variety of essential cardiovascular, metabolic, and immunological functions [12]. They can inhibit MAPK activities via the induction of MKP-1, inhibiting transcriptional and post-transcriptional mechanisms controlled by MAPK [13,14]. Glucocorticoids also regulate NF-kB in complex ways, involving transcription and translocation-regulated mechanisms [15,16,17]. It is recognised that glucocorticoids can significantly inhibit TGFβ production and modulate the activities of various members of the SMAD family and EndMT [18,19]. We and others have assessed the role of synthetic glucocorticoids (man-made) such as Dexamethasone in modulating vascular inflammation and IH in vein grafts [9,10,20,21].

In this study, we aim to gain insight into the effects of acute arterial haemodynamics on EndMT activation in HUVECs in vitro and LSVs ex vivo and explore Dexamethasone’s (Dex) role in regulating EndMT in LSVs.

## 2. Materials and Methods

### 2.1. Reagents and Antibodies

Pharmacological inhibitors of TGFβ (Selleckchem, Houston, TX, USA, S21, SB505124), Dexamethasone (Sigma-Aldrich, St. Louis, MO, USA, D2915), and antibodies were purchased commercially. The Appendix A includes a list of the antibodies used (Appendix A).

### 2.2. Comparative Reverse-Transcriptase Polymerase Chain Reaction

Total RNA was extracted using RNeasy mini kit (Qiagen, Venlo, The Netherlands, 74104). For mRNA studies, cDNA was synthesised from total RNA using the Tetro cDNA synthesis kit (Meridian Bioscience, Cincinnati, OH, USA, BIO-65043). Gene expression was determined by quantitative real-time polymerase chain reaction (qRT-PCR) using the SensiFast Probe Hi-ROX kit (Meridian Bioscience, Cincinnati, OH, USA, BIO-82005) and gene-specific primers on Rotor gene Q (Qiagen) using the manufacturer’s protocol. Relative levels were calculated using the 2^−(ΔΔCt)^ method, and mRNA expression was normalised to housekeeping gene PPIA. The primers list is included in the Appendix A.

### 2.3. Silencing of TWIST1 and TWIST2

HUVECs were transiently transfected at 80% confluency using Lipofectamine RNAiMAX (Invitrogen, Carlsbad, CA, USA, 13778075) transfection reagent and 30 μmol/L final concentration of TWIST1 siRNA (ThermoFisher Scientific, Waltham, MA, USA, s14523), TWIST2 siRNA (ThermoFisher Scientific, s42174), or Silencer™ Select Negative Control No. 1 siRNA (ThermoFisher Scientific, 4390843) in Opti-MEM reduced serum media (ThermoFisher Scientific, 31985070) according to the manufacturer’s protocol for 24 h. All in vitro experiments were conducted using HUVECs grown until passage 7.

### 2.4. EC Culture and Shear Stress

HUVECs were obtained from PromoCell (St. Louis, MO, USA, C-12203) and were cultured in Endothelial Cell Growth Medium 2 (PromoCell, C-22011) to full confluency on glass microscope slides precoated with 1% (*w*/*v*) gelatine (Sigma-Aldrich, G1393). HUVECs were then exposed to laminar, unidirectional shear stress (at 0.5 or 12 dyn/cm^2^ to simulate venous and arterial shear stress rates, respectively) for varying times, using parallel-plate flow chambers described previously or maintained in static conditions [10,11,22]. The glass slides were placed into the parallel-plate chambers and sealed with a silicon sheet gasket. A reservoir containing 30 mL RPMI 1640 culture medium (ThermoFisher Scientific 11875093), supplemented with 2% (*w*/*v*) FCS, 100 µg/mL penicillin, and 100 U/mL streptomycin attached to a closed-circuit loop of silicon tubing (VWR, Radnor, PA, USA and Elkay, Rotherham, UK), was connected to the chambers. HUVECs were then cultured at 37 °C and 5% CO_2_, and shear stress was applied using a multi-channel peristaltic pump (Watson-Marlow, Falmouth, UK). Shear stress rates were calculated for a slit die assuming the viscosity of water at 37 °C, using the following equation: τ = 6µǪ/wh^2^, where τ represents shear stress, µ represents viscosity, Ǫ represents flow rate, w represents width, and h represents height.

### 2.5. Ex Vivo Perfusion of Veins

We used our previously described model for ex vivo vessel perfusion [9,10]. Briefly, surplus segments of human LSVs, resected during surgery from anonymised, consenting patients, were placed immediately in RPMI 1640 culture medium with 10% (*w*/*v*) foetal calf serum (FCS), 100 µg/mL penicillin, and 100 U/mL streptomycin. The study was approved under the Leicester Biomedical Research Centre (BRICCS Ethics Ref: 09/H0406/114). Informed consent was obtained from all study participants before surgery, and our use of human tissue conformed to the principles outlined in the Declaration of Helsinki. Sections of 6 cm were cut for use in ex vivo perfusion. LSVs were either pretreated with Dex (10 µM/L for 60 min) or remained untreated. Vein sections were cannulated with a 1/16” size male luer fitting (World Precision Instruments, Sarasota, FL, USA) and secured with a fine surgical tie. They were then mounted on a perfusion apparatus and exposed to arterial haemodynamics for different times or were maintained under static conditions as a control. Veins attached to the flow system were perfused under a mean arterial pressure of 65 mm Hg with M199 medium (containing 20% FCS, 100 U/mL penicillin, and 0.1 mg/mL streptomycin), which was oxygenated (oxygen content, 20 mL/L) and prewarmed to 37 °C. We targeted a wall shear stress of 12 ± 0.2 dyn/cm^2^ and adjusted flow rates based on the diameter of the cannulae. Shear stress was calculated using the equation τ = 4µO˛/πr^3^, where τ represents shear stress, µ represents viscosity, O˛ represents flow rate, π represents pi, and r represents radius, assuming the flow is through a non-deformable cylinder and is laminar and considering the viscosity of water at 37 °C.

### 2.6. Detection of Proteins in Cultured Cells

Protein expression in cultured cells was measured by immunocytochemistry using the previously established method [10,11,22]. Briefly, HUVECs were cultured on glass slides (ThermoFisher Scientific, 13192131) until confluent, and then they were exposed to shear stress or maintained under static conditions and fixed in slides using pre-chilled 10% neutral buffered formalin (Sigma Aldrich, HT5014-1CS) for 15 min at 4 °C. Thereafter, slides were dehydrated in ethanol (50–100%) and allowed to dry. Cells were incubated in Immunofluorescence Blocking Buffer (Cell Signaling Technology, Danvers, MA, USA, 12411) for 30 min at room temperature, followed by overnight primary antibody incubation in Immunofluorescence Antibody Dilution Buffer (Cell Signaling Technology, 12378) at 4 °C and AlexaFluor fluorophore-conjugated secondary antibody for 1 h at room temperature. Immunoglobulin-matched controls at the same concentration as the primary antibodies were used as staining-specific negative controls. Nuclei were labelled with diamidino phenylindole (DAPI)-dilactate (Invitrogen) in PBS for 30 s, and coverslips were mounted with ProLong^®^ Gold Antifade Reagent (ThermoFisher Scientific, P36930). Slides were visualised using a Zeiss Axio Observer Z1 inverted microscope (Carl Zeiss, Oberkochen, Germany). Quantifying fluorescence intensity was measured as integrated intensity in whole cells divided by a total number of cells using CellProfiler 2.0.

Alternatively, levels of proteins of interest were measured in total cell lysates prepared using RIPA Lysis and Extraction Buffer (ThermoFisher Scientific, 89900). This was followed by western blotting using specific primary antibodies, horse radish peroxidase-conjugated secondary antibodies, and chemiluminescent detection as previously reported [22].

### 2.7. Immunohistochemistry

As previously reported, the expression levels of specific proteins were assessed in veins by immunohistochemistry [9,10,11]. Slides were incubated in pre-chilled 10% neutral buffered formalin (Sigma Aldrich, HT5014-1CS) for 15 min at 4 °C. Thereafter, slides were dehydrated in ethanol (50–100%) and allowed to dry. Slides were deparaffinised for staining on formalin-fixed, paraffin-embedded human samples and underwent antigen retrieval. Sections were then incubated in an Immunofluorescence Blocking Buffer (Cell Signalling Technology, 12411) for 30 min at room temperature, followed by overnight primary antibody incubation in an Immunofluorescence Antibody Dilution Buffer (Cell Signalling Technology, 12378) at 4 °C. A secondary antibody (HRP linked) was applied for one hour at room temperature, followed by nuclei staining with DAPI. Control slides were routinely stained in parallel by substituting IgG or the specific IgG isotype. All images were acquired using a Zeiss Axioscope with AxioVision V4.3 software or a Zeiss LSM 510 UV scanning confocal microscope (Carl Zeiss, GmBH, Oberkochen, Germany). As reported previously, the expression of proteins of interest was assessed by quantifying fluorescence intensity for multiple cells [9,11].

### 2.8. RNAscope In Situ Hybridisation

We used the previously reported method [10]. Briefly, frozen vein sections were processed for fluorescent in situ hybridisation by RNAscope according to the manufacturer’s guidelines (Bio-Techne Ltd., Abingdon, UK), as we described previously. Briefly, genes examined in the vein sections were TWIST1 (ACDBio, Newark, CA, USA, 470291), TWIST2 (ACDBio, and SNAI1 (ACDBio, 560421-C2), and hybridisation was performed using the RNAscope^®^ Multiplex Fluorescent Reagent Kit v2 (Bio-Techne Ltd., Abingdon, UK, 323100). These sections were co-stained with CD31, VE-Cadherin, or α-SMA antibodies using an RNAscope^®^ Multiplex Fluorescent v2 Assay combined with the Immunofluorescence-Integrated Co-Detection Workflow (Bio-Techne Ltd.). Slides were covered and slipped with ProLong™ Gold Antifade Mountant and stored at 4 °C in the dark before imaging. TWIST1&2 and SNAI1 puncta were counted selectively in CD31^+^ cells using the cell-counter plugin in Fiji [23]. After the RNAscope assay, dots were quantified based on the average number of dots per cell, as previously described [10,24].

### 2.9. LSV Culture Well

LSVs were cultured after 4 h of arterial flow exposure using the modified culture-well model as previously described. Briefly, the vein was placed in a wash medium (20 mm Hepes-buffered RPMI 1640 supplemented with 2 mm l-glutamine, 8 mg/mL gentamycin, 100 IU/mL penicillin, and 100 mg/mL streptomycin). Segments were opened longitudinally and cut transversely into three 5–10 mm segments. Vein segments were cultured separately with the endothelial surface uppermost, in culture medium (RPMI 1640 supplemented with 30% foetal calf serum, 2 mm L-glutamine, 100 IU/mL penicillin, and 100 mg/mL streptomycin) under an atmosphere of 95% air and 5% CO_2_ for 10 days with culture media being changed every 2 days.

### 2.10. Spatial Cell Sequencing

This was reported in detail previously [25]. Briefly, for sectioning in preparation for Visium, blocks were equilibrated to −18 °C, and 10 mm-thick sections were mounted onto the active sequencing areas of the 10× Genomics Visium slides. Hemotoxylin and eosin staining was performed according to the 10× Genomics Visium protocol [26]. Briefly, tissue sections were placed onto the Visium Gene Expression Slide, fixed with ice-cold methanol, stained with haematoxylin and eosin (H&E), and imaged using the Olympus FV1000 confocal microscope (Olympus, Tokyo, Japan). Tissue permeabilisation was performed according to an optimised protocol. Reverse transcription (RT) was undertaken on a thermocycler to produce spatially barcoded, full-length cDNA from poly-adenylated mRNA on the slide. Second-strand synthesis used the Second Strand Reagent, Primer, and Enzyme mix, followed by denaturation and transfer of the cDNA from each capture area to a corresponding tube for cDNA amplification and library construction (following cycle-number determination and quality control). Cycle-number determination was undertaken using the provided KAPA SYBR FAST qPCR Master Mix and cDNA Primers, whilst cDNA amplification was conducted using the provided Amp Mix and cDNA Primers. This resulted in a solution of spatially barcoded, full-length amplified cDNA generated by PCR for subsequent library construction. Final quality control was determined using the Agilent Bioanalyser High Sensitivity Kit (Santa Clara, CA, USA). Library construction was undertaken using a fixed proportion of the total cDNA (40 uL), with enzymatic fragmentation and size selection utilised to optimise the amplicon size. End-repair, A-tailing, adaptor ligation and PCR were utilised to add P5, P7, i7, and i5 sample indexes, along with a TruSeq Read 2 primer sequence. Libraries were sequenced using PE150 on the NovaSeq platform (Novogene, Beijing, China). Spatial sequencing data were processed using the SpaceRanger v1.3.1 pipeline, aligned to the associated 10X Genomics Human Genome Reference GRCh38-2020-A.

Brightfield images in high-resolution tiff format were manually aligned using 10× Loupe Browser 6.4.1 software to ensure accurate orientations of the expression data to the tissue image. Count and image data for all samples were imported into Seurat v4.0 for further analysis. Data were normalised using SCTransform, correcting for batch effects between different runs. All sample datasets were integrated for joint comparisons following the Seurat SCTransform integration workflow, with 3000 integration features and all common genes between samples [27]. Aggregated spot expression data from all samples were tested with muscatv1.6 in R v4.0.0 to better account for spot variation in a multi-sample, multi-batch, multi-state experiment with group replicates [28]. Unsupervised clustering was performed using the Seurat FindClusters function, and cluster identity was inferred using a combination OF unique gene marker identification and analysis of the top 30 most highly expressed genes in each grouping. Pathway analysis was undertaken for samples reaching significance (*p* < 0.05 adjusted) using the gprofiler package [29] and associated public datasets (KEGG, REAC, Wikipathways, Gene Ontology, biological processes, cellular components, and molecular functions [30,31,32,33]. Network analysis was performed using the Cytoscape application (version 3.9.1) [29]. Connections were inferred between genes of interest by running a heat diffusion analysis with standard parameters [34]. Global network analysis was subsequently undertaken using default confidence and interaction parameters. Genes were visually quantified using a colour gradient scale, ranging from red to blue, based on their LogFC value. Specific genes of interest were identified and subsetted from the global network to identify the nearest neighbouring interactors from known pathways. This demonstrated and predicted a significant overlap and connections between these genes and pathways involved using diffusion analysis [34].

### 2.11. Statistics

Data were subjected to a paired, two-tailed *t*-test for experiments where only two groups were analysed. For experiments where more than two groups were analysed, a one- or two-way ANOVA was used depending on the number of independent variables, followed by post hoc pairwise comparisons with Bonferroni correction for multiple comparisons. If datasets were large enough (for example, for immunocytochemical analyses, where 20 images per sample were analysed), normal distribution was assessed with the D’Agostino–Pearson test; all data assessed passed normality tests, so parametric analyses were appropriate. The cut-off value for statistical significance was 0.05. Data are presented as mean ± SD from 4 independent experiments for all presented data unless indicated otherwise. Statistical analysis was performed with GraphPad Prism 7.0. The corresponding author had full access to all the data in the study and takes responsibility for its integrity and the data analysis.

## 3. Results

### 3.1. Acute Arterial Shear Stress Promotes EndMT in Venous ECs In Vitro

We investigated the impact of acute laminar arterial shear stress (LSS) on the activation of EndMT in vitro. We observed that applying LSS for 4 h activated inflammatory mediators in HUVECs, such as increased expression of MCP-1 and CXCL8, as previously reported [9,11]. We also noted a significant downregulation of VE-CADHERIN (Vecad) and upregulation of VIMENTIN (VIM), TWIST1, and TWIST2 gene expression (Figure 1A). CD31 and SMA showed similar trends but were not statistically significant (Appendix A). These data suggest that ASS can trigger EndMT in HUVECs. To examine the role of TGFβ, we used a pharmacological inhibitor of TGFβ that is known to inhibit activin A receptor type II-like kinase (ALK5) (SB505124) [35]. We observed that the pretreatment of HUVECs with the inhibitor (5 μmol/L for 60 min) suppressed the induction of the phenotype changes noted above both at transcript (Figure 1B) and protein levels at 4 h (Figure 1C). This suppression was associated with an early (45 min) suppression of ALK5 activation and SMAD 2/3 phosphorylation (Figure 1C). Thus, we conclude that HUVECs are sensitive to TGFβ-dependent induction of EndMT by LSS.

Next, we assessed the role of TWIST1&2 using specific siRNAs that suppressed messenger RNA expression in sheared HUVECs (Appendix A). We observed that the silencing of TWIST1 or TWIST2 suppressed EndMT changes HUVECs under LSS at transcript levels (Figure 1D) and protein levels using immunostaining (Figure 1E) and WB (Appendix A) but did not affect upstream ALK5 activation of SMAD 2/3 phosphorylation (Appendix A), suggesting that both TWIST1 and TWIST2 can play a role in the EndMT process under LSS in vitro. Although TWIST1 and TWIST2 did not coregulate each other’s expression, both are required to cooperate under acute arterial haemodynamics to activate responses as has been shown under different conditions [36].

### 3.2. Dexamethasone Pretreatment Suppressed EndMT in Venous Ecs

Using a pretreatment dose of Dex of 10 μmol/L for 60 min (based on our previous work) [9,10], we noted significant alteration in HUVEC responses to LSS. The pretreatment with Dex resulted in the suppression of ALK5 activation and SMAD2/3 phosphorylation at 45 min (Figure 2A). Moreover, it was associated with the suppression of TWIST1, TWIST2, and VIM while restoring the expression of Vecad both at transcript (Figure 2B) and protein levels using immunostaining (Figure 2C) and WB (Appendix A). These results suggest that the use of Dex can suppress EndMT changes in response to LSS in HUVECs in part by targeting the ALK5/SMAD/TWIST pathway.

### 3.3. EndMT Activation in Veins Exposed to Arterial Flow Ex Vivo Using Untargeted Spatial Cell Sequencing

We validated our in vitro findings by studying freshly harvested human saphenous veins exposed to acute arterial flow ex vivo. In a series of first-in-human experiments (*n* = 4), we recently performed spatial cell RNA sequencing on vein segments exposed to acute arterial haemodynamics for 4 h [25]. We identified a cluster of cells with both EC and SMC phenotypes (EC/SMC) (Figure 3A,B), suggesting that it may represent an intermediate EndMT, as reported previously [24]. In this cluster, 16954 genes were identified (Figure 3C). Of these, 486 were significantly differentially regulated under flow conditions, including TWIST2 (*p* = 0.028) (Appendix A). A list of all the significantly regulated genes is included in the Appendix A. Global pathway analysis showed correlations made to the different groupings, including the regulation of signalling, specifically the inflammatory and immune responses (cytokine production and responses), leukocyte activation, migration, differentiation, adhesion, wound healing responses, and activation/regulation of the p53-MAPK signalling cascade. Additionally, there was a correlation to the regulation of cell behaviour, including activation, proliferation, differentiation, migration, cell–cell communication, adhesion, and motility, with specific reference to smooth muscle-cell proliferation and migration (Figure 3D). Global network analysis identified a total of 2204 connections between 436 of the 519 differentially expressed genes in the EC/SMC cluster (Appendix A). When focusing on TWIST2 network analysis in this cluster specifically, we noted associations with genes such as TGIF1, FOXC2, CD44, CYP1B1, NFIC, REL, and FOXC1 and their subsequent associations. Such genes are involved in processes such as inflammation, EndMT, immune signalling, differentiation, and apoptosis [37,38,39,40,41,42,43,44] (Figure 3E).

### 3.4. Dexamethasone Pretreatment Suppressed EndMT in Veins Exposed to Arterial Flow Ex Vivo

We next set up, in order to validate the above finding in relation to genes associated with EndMT, targeted approaches including RT-PCR for the whole tissue and RNAscope that allows the localisation of genes of interest’s expression in the vein sections (considering the limitation of spatial sequencing in terms of single-cell localisation) and studied the impact of a brief pretreatment of veins with Dex ex- vivo (other validated genes of interest in Appendix A). In addition to suppressing proinflammatory responses to acute arterial flow (Appendix A), Dex pretreatment suppressed TWIST1 and TWIST2 expression at transcript levels using RT-PCR (Figure 4A) and RNAscope at 4 h, which demonstrated the activation and localisation of both transcriptions of RNA in the ECs, which were suppressed by Dex treatment (Figure 4B). Additionally, this brief pretreatment was able to suppress EndMT phenotype changes in ECs for VEcad and VIM at RNA (Figure 4C). Moreover, Dex pretreatment significantly modulated TWIST1 and TWIST2, VEcad, and VIM protein levels, which was observed using immunohistostaining (Figure 4D). Furthermore, immunohistostaining revealed that Dex action was also associated with a rapid (45 min) significant reduction in ALK5 expression and SMAD 2/3 activation (phosphorylation) (Figure 4E). Interestingly, when we looked at SNAIL1 transcription using RNAscope, we noted no activation in ECs and only presence in SMCs, which was suppressed by Dex pretreatment (Appendix A), suggesting the TWIST and SNAIL transcription factors may work on specific cell types. Moreover, considering the limitation of running long-timescale flow experiments, we compared the effect of brief Dex pretreatment followed by LSV culture for 10 days. We noted that the brief single treatment was sufficient for suppressing EndMT at 10 days post-culture (Appendix A). Interestingly, when we compared the role of low flow on the activation of LSV ECs, we noted that low flow did not trigger any EndMT changes in LSVs ex vivo (Appendix A). Additionally, no changes in total SMAD levels were noted (Appendix A). Thus, we conclude that a brief pretreatment with Dex (60 min) can suppress EndMT activation in LSV ECs in response to shear stress by suppressing the TGFβ/SMAD1 pathway.

## 4. Discussion

Veins grafted into the arterial circulation are exposed to an acute increase in blood flow. It is recognised that haemodynamics can influence the inflammatory process by controlling leukocyte margination and adhesive interactions and generating shear stress, which alters EC physiology [3,4,5]. We have previously shown that venous and arterial ECs respond differently to shear stress, suggesting that an interaction between local haemodynamics and stable vessel-specific EC phenotypes regulates the susceptibility of veins and arteries to the inflammatory process [9]. We also demonstrated that high but not low shear stress is associated with such proinflammatory responses in venous ECs [11]. In this study, we looked at the impact of exposing venous ECs/veins to arterial haemodynamics on the activation of EndMT and the pathway involved. Many factors can stimulate EndMT; the most common are the TGFβ and BMP growth factors [45,46,47]. TGFβ1 binds to a complex of receptors that include type II receptor (TβRII) and type I receptor activin-like kinase 5 (ALK5), which promotes signalling through SMAD2/3, [48], whereas BMP2 and BMP4 primarily signal through the ALK2 receptor [49,50].

Since the presence of TGFβ in the neointima of animal models was reported [51], experimental studies have shown that upregulation of TGFβ or the addition of exogenous TGFβ resulted in increased intimal thickening or neointima formation [52,53,54]. Many groups have demonstrated that TGFβ1 is upregulated early after injury [55,56,57]. Such upregulation can separately activate both SMAD1/5- and SMAD2-mediated pathways, which can play an essential role in developing IH in vein grafts [58].

Cooley et al., using in vivo murine cell lineage-tracing models, showed that endothelial-derived cells contribute to neointimal formation through EndMT, which is dependent upon early activation of the SMAD2/3-Slug signalling pathway [8]. Histological examination of post-mortem human vein graft tissue corroborated the changes observed in our mouse-vein graft model, suggesting that EndMT functions during human-vein graft remodelling.

Our work supports and expands on what has been previously reported. We first examined the effects of flow on the expression of the EndMT markers in vitro. We observed that acute exposure to high shear stress was associated with rapid activation of ALK5, pSMAD2/3; activation of TWIST2 and TWIST1 transcription factors; and changes in EC-supporting phenotype changes. We confirmed that ALK5 mediates EndMT by using a specific inhibitor that could suppress the impact of acute shear on ECs. Next, we showed that silencing TWIST1 and TWIST2 transcription factors can suppress EndMT changes in ECs, confirming their role. Our previous work showed that a brief pretreatment of ECs with Dex can suppress the activation of EndMT in vitro. Next, we performed spatial cell sequencing of LSV sections exposed to arterial flow. We showed the activation of different proinflammatory mediators and TWIST1&2 in response to acute arterial flow based on clustering, which identified a subset of cells with both EC and SMC characteristics. In this cluster, we noted that TWIST2 was significantly activated by flow, suggesting that these cells are at an intermediate stage of EndMT or what has been described previously as the hybrid cells expressing both EC- and SMC-specific molecular markers and most likely representing an intermediary phenotype between endothelial cells and mesenchymal/myofibroblastic cells [6]. We validated these untargeted results ex vivo using RT-QPCR, RNAscope, and immunohistostaining. This showed that the exposure of LSVs to acute arterial flow ex vivo can trigger EndMT in ECs. Moreover, we demonstrated that a brief pretreatment of LSVs with Dex can suppress these changes.

The oral administration of a non-specific inhibitor of TGFβ biosynthesis in patients after PCI failed to improve their conditions significantly. It was associated with substantial adverse effects [59], considering that the TGFβ-signalling pathway is complex and mediates several functions in other organs. Thus, a more attractive alternative is the ability to use local/topical therapeutics that can target pathways in specific tissues without exposing patients to the adverse effects of systemic usage. Glucocorticoids inhibit TGFβ production and modulate the activities of different SMAD family members and EndMT members [19,60,61]. Here, we prove that brief pretreatment with Dex can prevent veins from exhibiting an inflammatory phenotype in response to arterial shear stress. Our findings suggest that the known beneficial effects of Dex treatment in the prevention of intimal hyperplasia may involve suppression of TGFβ/SMAD EndMT in addition to suppressing vascular inflammation via the MAPK pathway and the regulation of OPN/microcalcification, as shown previously [9,10].

Although we focused on the ALK5/SMAD pathway in this study, evidence suggests that glucocorticoids effectively inhibit bone morphogenetic protein expression such as BMP2 [62]. Thus, it will be interesting to address EndMT in LSVs and the role of BMPs in future studies. Addressing the role of NF-kB in triggering EndMT via the activation of transcription factors such as TWIST1&2 will be very interesting, considering that spatial data network analysis and the links between NF-kB, inflammation, and TWIST2 will also add insight into this complex phenomenon. Most importantly, this study focuses on short-term changes and will be of great value in the future regarding expansion on previously conducted small-animal studies to investigate the impact of short topical pretreatment with Dexamethasone on IH formation in a large-animal model, incorporating novel techniques such as spatial cell sequencing and RNASCope.

## 5. Conclusions

Our results demonstrated that TGFβ-/SMAD-/TWIST1&2-mediated EndMT is activated within vein grafts in response to acute exposure to arterial haemodynamics ex vivo and that a brief pretreatment of veins with Dexamethasone can suppress both its activation and associated microcalcification, which may modulate the development of intimal hyperplasia in vein grafts.

## Figures and Tables

**Figure 1 cells-14-01369-f001:**
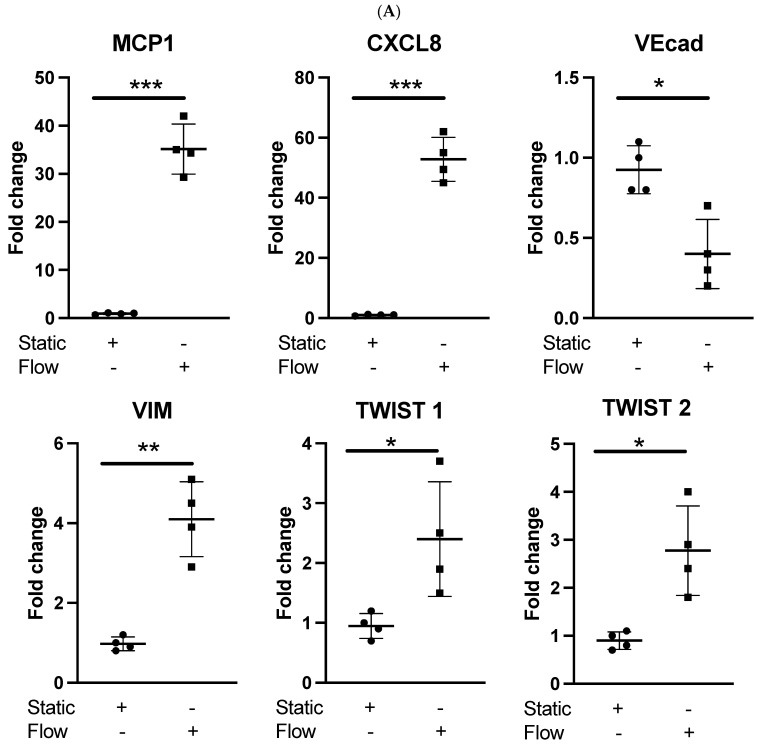
Effects of arterial shear stress on proinflammatory and EndMT genes in venous endothelial cells (Ecs). (**A**) HUVECs were cultured under static conditions or exposed to LSS for 4 h. Transcript levels of MCP1, CXCL8, Vecad, VIM, TWIST1, and TWIST2 were measured using comparative RT-PCR. Results from four independent experiments are presented. (**B**) HUVECs were pretreated with an ALK5 inhibitor (5 μmol/L) or vehicle alone for 60 min then exposed to LSS for either 45 min or 4 h or cultured under static conditions. The transcript levels of Vecad, VIM, TWIST1, and TWIST2 were measured using comparative RT-PCR. Data from four independent experiments are shown. (**C**) HUVECs were pretreated with an ALK5 inhibitor or vehicle alone for 60 min, followed by exposure to LSS for 4 h or static culture. Expression levels of TWIST1, TWIST2, VIM, and Vecad were assessed at 4 h. ALK5 and P-SMAD 2/3 levels were evaluated at 45 min through immunofluorescence staining with specific antibodies. The results were quantified in multiple Ecs and averaged across each experimental group. Representative images and mean values from four independent experiments are displayed. (**D**) HUVECs were transfected with TWIST1- or TWIST2-specific siRNA or scrambled control (Scr). Cells were exposed to LSS or cultured under static conditions for 4 h. Vecad, VIM, TWIST1, and TWIST2 transcript levels were measured using comparative RT-PCR. Mean values from four independent experiments are presented. (**E**) Expression levels of TWIST1, TWIST2, VIM, and Vecad were assessed at 4 h in HUVECs that were transfected with TWIST1- or TWIST2-specific siRNA or scrambled control (Scr) after exposure to LSS or static culture for 4 h. The results were quantified in multiple Ecs and averaged across each experimental group. Representative images and mean values from four independent experiments are presented (‘+’ indicates the presence and ‘−’ the absence of the indicated experimental condition) (* *p* < 0.05; ** *p* < 0.01; *** *p* < 0.001).

**Figure 2 cells-14-01369-f002:**
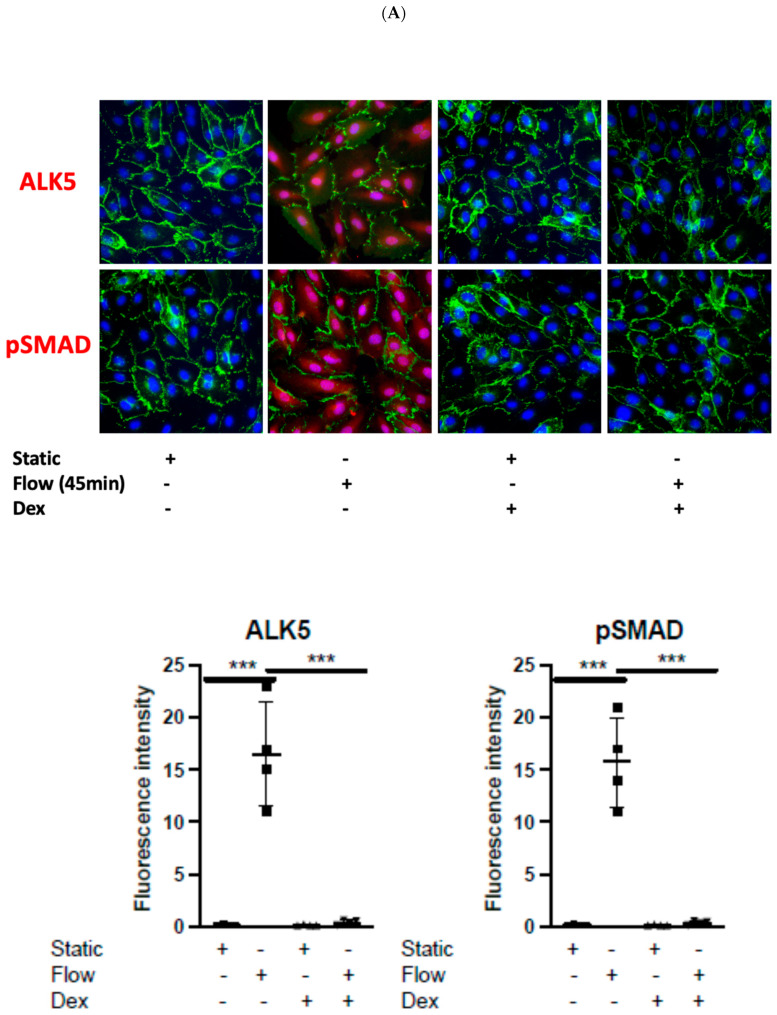
Dexamethasone suppressed EndMT activation by shear stress in cultured venous ECs. (**A**) HUVECs were pretreated with Dexamethasone (10 μmol/L for 60 min) or left untreated. They were then exposed to LSS for 45 min. ALK5 and phosphorylated SMAD 2/3 were assessed using immunofluorescence staining with specific antibodies. Representative images are shown, and values from four independent experiments are presented. (**B**) HUVECs were similarly pretreated with Dexamethasone or remained untreated, followed by exposure to LSS for 4 h. RT-PCR was used to measure transcript levels of TWIST1, TWIST2, VEcad, and VIM. (**C**) The expression levels of TWIST1, TWIST2, VIM, and VEcad were evaluated at 4 h in HUVECs pretreated with Dexamethasone or left untreated. This was conducted using immunofluorescence staining with specific antibodies. The results were quantified in multiple ECs. Representative images are shown, and values from four independent experiments are presented (‘+’ indicates the presence and ‘−’ the absence of the indicated experimental condition) (* *p* < 0.05; ** *p* < 0.01; *** *p* < 0.001).

**Figure 3 cells-14-01369-f003:**
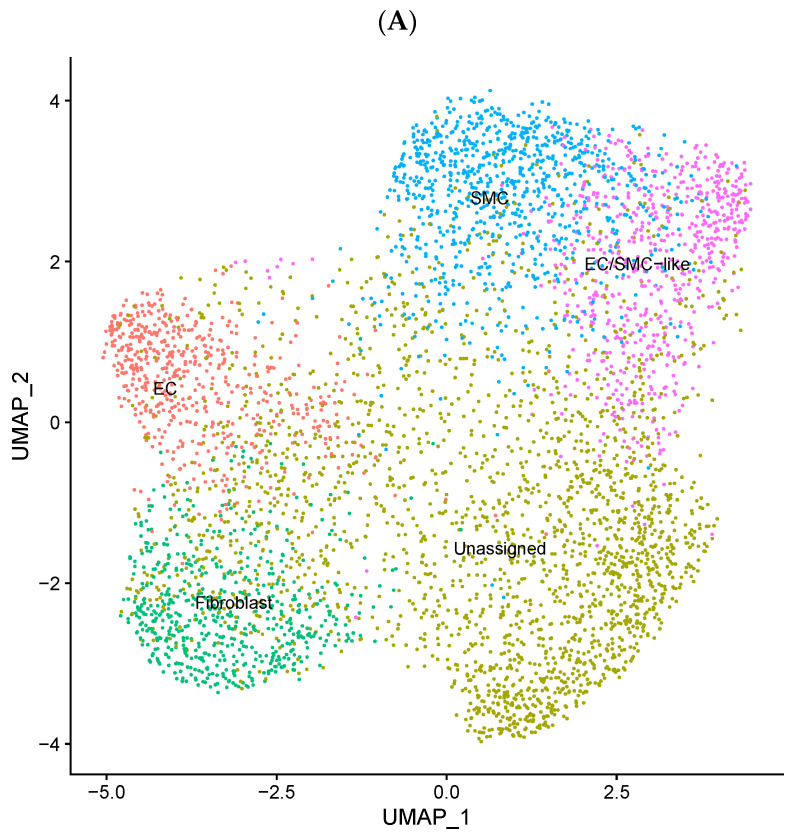
EndMT activation in veins exposed to arterial shear stress ex vivo using untargeted spatial cell sequencing. LSV segments were mounted on a perfusion apparatus and exposed to arterial flow for 4 h or maintained under static conditions as a control. (**A**) A cluster of cells demonstrating both endothelial cell (EC) and smooth muscle-cell (SMC) phenotypes (EC/SMC). (**B**) Representative images of a lower limb venous system (LSV) tissue section (10 μm) microscopy image, along with a 10X Visium spot coverage grid overlay illustrating the spatial distribution of the EC/SMC cluster. (**C**) A volcano plot of the identified EC/SMC cluster, displaying all differentially expressed genes (DEGs) between control samples and those exposed to 4 h of arterial haemodynamics (n = 4). (**D**) Global pathway analysis of the EC/SMC cluster, highlighting all significant differentially expressed genes (adjusted *p*-value < 0.05), with the top 30 connections presented. Enrichment is scaled based on significance (−log10 *p*-value) and visually ranked by intersection size (overlap of DEGs with enriched pathways). (**E**) A network of significant differentially expressed genes (adjusted *p*-value < 0.05) in the TWIST2 subset of the EC/SMC cluster in samples exposed to ex vivo arterial haemodynamic conditions.

**Figure 4 cells-14-01369-f004:**
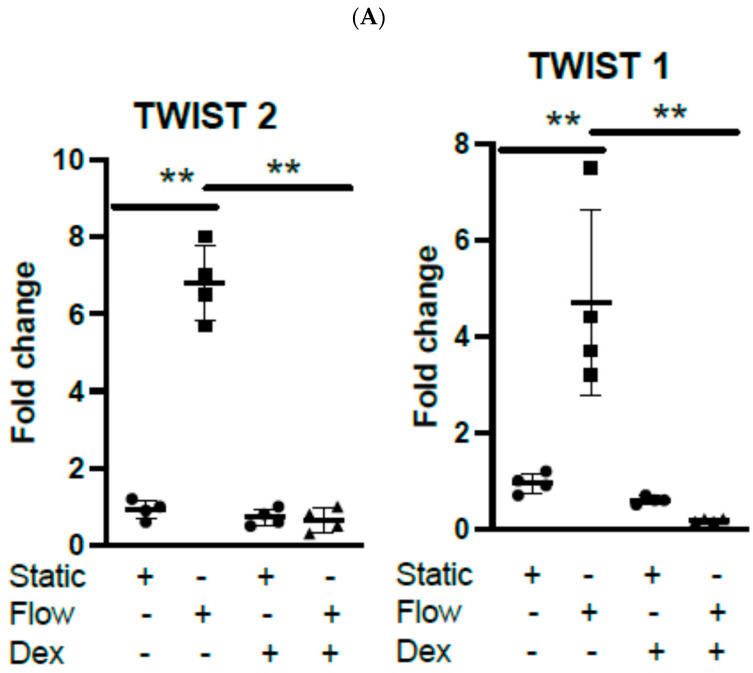
Pretreatment of human long saphenous veins with Dexamethasone suppressed EndMT activation by shear stress in endothelial cells (ECs). (**A**) LSV segments were pretreated with Dexamethasone (10 μmol/L for 60 min) or remained untreated. Following this, the veins were mounted on a perfusion apparatus and exposed to arterial flow for 4 h or maintained under static conditions as a control. The transcript levels of TWIST2 and TWIST1 in the whole tissue were measured using comparative RT-PCR. Results are shown as values from four independent experiments, with mean values presented. (**B**) The expression levels of TWIST1 and TWIST2 at 4 h were assessed using RNAscope with probes specific for TWIST1 and TWIST2 genes (indicated by arrows showing gene expression). This was quantified across multiple ECs. Representative images, values from four independent experiments, and mean values are shown. (**C**) LSV segments were either pretreated with Dexamethasone (10 μmol/L for 60 min) or remained untreated, and then they were mounted on a perfusion apparatus and exposed to arterial flow for 4 h or kept under static conditions. The transcript levels of VE-cadherin (VEcad) and vimentin (VIM) in the whole tissue were measured using comparative RT-PCR. Results are from four independent experiments with mean values presented. (**D**) TWIST1&2, VIM, and VEcad expression levels were assessed at 4 h via immunofluorescence staining with specific antibodies. This was quantified across multiple ECs and averaged for each experimental group. Representative images, values from four independent experiments, and mean values are shown. (**E**) ALK5 and P-SMAD 2/3 expression levels were assessed at 45 min using immunofluorescence staining with specific antibodies. Quantification was performed in multiple ECs and averaged for each experimental group. Representative images, values from four independent experiments, and mean values are shown (‘+’ indicates the presence and ‘−’ the absence of the indicated experimental condition) (* *p* < 0.05; ** *p* < 0.01; *** *p* < 0.001).

## Data Availability

The raw data supporting the conclusions of this article will be made available by the authors on request.

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
