# Peer review of "Acute Shear Stress Induces TWIST-Mediated EndMT in Venous Endothelial Cells and Human Long Saphenous Veins"

_cells, 2025, doi:10.3390/cells14171369_

Round 1

Reviewer 1 Report

Comments and Suggestions for Authors

The manuscript by Ladak et al entitled “Acute shear stress induces TWIST-mediated EndMT in venous EC and human long saphenous veins" aims to interrogate the role of hemodynamic changes on Tgfb-driven EndoMT. This study used HUVEC in vitro and human long saphenous veins ex vivo exposed to static or LSS conditions in the presence or absence of a pharmacological inhibitor of Tgfb signaling and reported a role of ALK5/Smad/TWIST1/2 axis in EndoMT transition in response to LSS. Overall, this is an interesting paper; the study was well-designed and executed.  However, there are a few concerns as outlined below:

  1. All protein expression data is solely based on IF studies. Several of those results should be validated using IB as a complementary approach.
  2. ALK5 and pSMAD expression (IF) in response to TGFb inhibitor and Dex were presented for 45 mins exposure to LSS while rest of the data (TWIST, VIM, VEcad, etc) were studied following 4h exposure to LSS. Is suppression of TGFb signaling sustained in cells exposed to LSS for 4h?  Data confirming suppression of ALK5 and pSMAD in 4h LSS-treated samples is needed.  Triple IF staining depicting Vim/VEcad/Alk5 or pSMAD and Twist/Alk5 or pSMAD is needed.
  3. In Fig 1, Twist siRNA is shown to block Flow-induced Vim mRNA expression. However, IB data (Suppl. Fig 1C) particularly with Twist 1 siRNA is not very convincing.  Moreover, many of those immunoblots are of poor quality.  Although protein expression was demonstrated using IF, those findings must be validated using IB. Likewise, Suppl. Fig 1B shows Twist mRNA levels in response to Twist siRNA gene silencing.  However, no protein expression is shown. Twist protein levels must be assessed to validate Twist deletion.
  4. Only pSmad expressions are presented. Where is the corresponding total Smad data?  For both IF and IB, total Smad levels are needed.
  5. Major Conclusion of this paper that Alk5/Smad pathway modulates EndoMT in response to LSS is solely based on the use of a pharmacological inhibitor of Tgfb signaling. These findings or at least some of them should be confirmed in cells or vein segments treated with Alk5/Smad gene silencing.
Comments on the Quality of English Language

There are a few typos, grammatical errors, and inconsistencies with the use of abbreviations that must be addressed.

Author Response

Comment 1 : All protein expression data is solely based on IF studies. Several of those results should be validated using IB as a complementary approach.

Answer 1: IF is a valid method to investigate protein expression as demonstrated with different studies and publications. We have included multiple WB figures to support the IF in the supp data. Additionally, we have also done total smad WB which is now included in the supp data.

Comment 2: ALK5 and pSMAD expression (IF) in response to TGFb inhibitor and Dex were presented for 45 mins exposure to LSS while rest of the data (TWIST, VIM, VEcad, etc) were studied following 4h exposure to LSS. Is suppression of TGFb signalling sustained in cells exposed to LSS for 4h?  Data confirming suppression of ALK5 and pSMAD in 4h LSS-treated samples is needed.  Triple IF staining depicting Vim/VEcad/Alk5 or pSMAD and Twist/Alk5 or pSMAD is needed.

Answer 2: Thank you for the comment. The reason is that we are investigating a signalling pathway that needs to be active before the expression of proteins. Based on our previous work, which we have cited, we know that signalling is expressed earlier and once the pathway is active then genes are expressed at a later stage and will be associated with corresponding protein expression. This was demonstrated in different previous studies by us and others which are cited in this manuscript. We have used a single point of 45 minutes that matches our previous work. Different short times, such as 30 or 60 minutes, can be considered, but this will require a large amount of work, and we don't feel it will contribute to the study, as we are not looking at temporal changes in signalling in this study.  We don't agree that we need to prove the sustained expression of the signalling pathway at 4 hours, as once the signalling is triggered, then the process starts. We would look at later activation of psmad, etc, if we are aiming to look at prolonged expression of VCAD, etc, which is not the scope of this study; thus, triple staining is not going to add to this study.

Comment 3: In Fig 1, Twist siRNA is shown to block Flow-induced Vim mRNA expression. However, IB data (Suppl. Fig 1C) particularly with Twist 1 siRNA is not very convincing.  Moreover, many of those immunoblots are of poor quality.  Although protein expression was demonstrated using IF, those findings must be validated using IB. Likewise, Suppl. Fig 1B shows Twist mRNA levels in response to Twist siRNA gene silencing.  However, no protein expression is shown. Twist protein levels must be assessed to validate Twist deletion.

Answer 3: We appreciate the comment, and these blots are extremely difficult to obtain, considering the tissue we are working with and that such proteins are not present in abundance in the cells. We have added them as supp supporting data to our well-validated IF studies. We don't agree that once silencing genes using a well-validated method, we need to demonstrate the absence of protein. We have shown in the study that the silencing of the gene will result in functional changes in response to stimulation.

Comment 4: Only pSmad expressions are presented. Where is the corresponding total Smad data?  For both IF and IB, total Smad levels are needed.

Answer 4: Thank you for the comment. We have added IF and WB of total SMAD to the supp data.

Comment 5: The Major Conclusion of this paper that the Alk5/Smad pathway modulates EndoMT in response to LSS is solely based on the use of a pharmacological inhibitor of Tgfb signalling. These findings or at least some of them should be confirmed in cells or vein segments treated with Alk5/Smad gene silencing.

Answer 5: The significant finding of this paper is that the treatment of Dex will inhibit EndMT triggered by acute high shear stress in veins. We used pharmacological inhibitors in addition to Dex as well as gene silencing, to look at pathways involved and to study the mechanisms, but the aim was to show that veins exposed to shear actually can develop EndMT and that Dex pretreatment can inhibit this. The focus was not to study ALK5/SMAD but rather TWIST and EndMT which we did in this paper. Silencing genes in vein segment tissue is not an easy task and requires a large amount of resources and is beyond the scope of this paper.

Reviewer 2 Report

Comments and Suggestions for Authors

Comments

Dear authors here are my suggestions:

Page 2, line 42: “arterial shear stress is associated with activating different proinflammatory mediators” you could mentioned more details and described your previous jobs.

Page 2, line 49: “Glucocorticoids” I believe better to write “glucocorticoids”

Page 2, line 51: “synthetic glucocorticoids” What do you mean synthetic? Explain please

Page 2, line 54: “in-vitro” and “ex-vivo” italic please. Continue in the whole text

Page 2, line 76: “hours” please write with the abbreviation h. Continue in the text

Page 2, line 87: “100μg/ml”, “100U/ml” correct please to mL. Continue in the text

Page 3, line 114: You could prepare shape/figure to explain the stress

Page 3, line 122: “minutes” use please the abbreviation min

Page 3, line 128: “seconds” correct please s

Page 4, line 155: “THE” correct please the letters, not in capital

Page 4, line 176: “For” correct please the letter “F”

Page 22, line 435: “a brief pretreatment with Dex” you should refer the time of pretretment with Dex. But you mentioned in different places in the text.

Page 27, line 521: “the activation OF” correct please OF

Page 29, line 596: Correct the style of reference 5. The same for references 12, 18, 19, 23, 28, 32, 33, 44, 45, 46, 47, 49, 57

Author Response

Comment 1:Page 2, line 42: “arterial shear stress is associated with activating different proinflammatory mediators” you could mentioned more details and described your previous jobs.

Answer 1: Thank you, we have expanded on that.

Comment 2:Page 2, line 49: “Glucocorticoids” I believe better to write “glucocorticoids”

Answer2: Thank you , corrected.

Comment 3: Page 2, line 51: “synthetic glucocorticoids” What do you mean synthetic? Explain please.

Answer 3: this means human made such as dex. we added this to the paper.

Commnet 4:Page 2, line 54: “in-vitro” and “ex-vivo” italic please. Continue in the whole text

Answer 4: corrected

Commnet 5:Page 2, line 76: “hours” please write with the abbreviation h. Continue in the text

Answer 5: changed.

Comment 6: Page 2, line 87: “100μg/ml”, “100U/ml” correct please to mL. Continue in the text

Answer 6: corrected

Commnet 7: Page 3, line 114: You could prepare shape/figure to explain the stress.

nswer 7: thank you, we are not clear what this means, we have provided an infographic for the paper

Comment 8: Page 3, line 122: “minutes” use please the abbreviation min

Answer 8: changed 

Comment 9: Page 3, line 128: “seconds” correct please s

Answer 9: changed 

Comment 10: Page 4, line 155: “THE” correct please the letters, not in capital

Answer 10: changed

Comment 11:Page 4, line 176: “For” correct please the letter “F”

Answer 11: corrected

Comment 12:Page 22, line 435: “a brief pretreatment with Dex” you should refer the time of pretretment with Dex. But you mentioned in different places in the text.

Answer 12: Added

Comment 13:Page 27, line 521: “the activation OF” correct please OF

Answer 13: corrected

Comment 14:Page 29, line 596: Correct the style of reference 5. The same for references 12, 18, 19, 23, 28, 32, 33, 44, 45, 46, 47, 49, 57

Answer 14: all styles corrected with Endnotes

Reviewer 3 Report

Comments and Suggestions for Authors

The study seems to be interesting. The authors analyzed primary vanous cells (HUVECs) in 4h shear stress conditions and found pro-inflammatory and EndMT changes that have been suppressed by pretretament with Dex. Furthermore, they observed similar changes in isolated saphenous vein subjected to the similar treatment.

However, something is still unclear for me.

Line 245: 
How did the authors get to the Twist1 and 2 molecules? Was there any kind of profiling performed first? There is no reference to their previous publication or the rationale, which means this has to be explained here. There are many transcription factors involved in EndMT start and progression, like, for example, Slug, Snail, FoxC2, Klf8 (and other KLFs), Zeb1, Zeb2, GSC. Have these been checked? I noticed that the authors have analyzed Snail1 expression and identified it specifically expressed and suppressed (by Dex) in SMC but not in ECs (lines 426 - 428). Do the authors consider the possibility that SMC expressing Snail1 represent those EndMT-converted former ECs that the authors themselves indicated as EC/SMC-like cells (in Fig.3A)? In such a case Snail1 expression would certainly mark EndMT cells whereas Twist1/2 would be unrelated or indicate an earlier step in EndMT conversion process (even though shRNA for Twist showed some effect)?! The same question is about FoxC2, which is indicated as significantly changed in Supplementary Table 3. Do the authors see alterations of FoxC2 expression in HUVECs under 4h shear stress?

General comment and question:
For me personally it is a bit difficult to accept the authors data that 4h of a stimulus (shear stress) is enough even for the new protein expression (Vim IFL data on Fig.1). To convert HUVEC into EndMT-like cells I used minimum 24h of a treatment (silencing of some gene - data unpublished). Therefore, my question is: How stable are those EndMT-related changes in cultured HUVECs, observed after 4h of shear stress? If to remove shear stress after 4h and change to static conditions do the cells revert the expression of those markers (Twist, Vim, VEC, Alk5, pSMAD RNA and protein) to the "baseline" level?

When the authors observe overexpression or suppression of a few genes (Vim, Twist, VEcad) does it mean that the real EndMT occurs? Can it be just a few EndMT-related genes (and maybe thousand of unreleated genes, which authors did not analyzed)? HUVECs undergone EndMT might change their morphology (from cobblestone to spindle-shape) and motility. Did the authors observed these effects? The more comprehensive assessment of the cells after the treatment can provide better characterization of the system.

Fig.S1B,E:
Silencing of Twist1 and Twist2 is only 50%! It is a bit difficult to believe that so small suppression level of 50% is enough to fully revert the effect of shear stress on Vim and VEcad. And these siRNA for Twist1 and Twist2 are used separately! How is it possible? If you inhibit just 50% of, for example, Twist1 (Fig.S1B), then full Twist2 is still available and also the remaining 50% of Twist1. And vice versa, when you inhibit Twist2 (only 50% --> Fig.S1B). Did the authors try to inhibit simultaneously both Twist 1 and 2? 

General question for the cell experiments and for saphenous vein:
Do the authors link proinflammatory changes and EndMT? One induces the other or they are parallel processes? Proinflammatory changes might be more relevant because they directly induce attraction conditions for the inflammatory cells, which leads to pathological consequences in the wall of a vein (or artery)? I think it needs to be carefully discussed, as I, for example, am not so sure if the EndMT-related changes are not just a secondary ones, while the proinflammatory changes might be the utmost important. Did the authors try to analyze cytokine profiles in HUVECs and LSV tissues after shear stress?

Minor:
Figure panels' placement in PDF is really weird. For example, Figure 1 panels are placed on various pages that it is really problematic to find a necessary page with the panel.

Author Response

Comment 1: How did the authors get to the Twist1 and 2 molecules? Was there any kind of profiling performed first? There is no reference to their previous publication or the rationale, which means this has to be explained here. There are many transcription factors involved in EndMT start and progression, like, for example, Slug, Snail, FoxC2, Klf8 (and other KLFs), Zeb1, Zeb2, GSC. Have these been checked? I noticed that the authors have analyzed Snail1 expression and identified it specifically expressed and suppressed (by Dex) in SMC but not in ECs (lines 426 - 428). Do the authors consider the possibility that SMC expressing Snail1 represent those EndMT-converted former ECs that the authors themselves indicated as EC/SMC-like cells (in Fig.3A)? In such a case Snail1 expression would certainly mark EndMT cells whereas Twist1/2 would be unrelated or indicate an earlier step in EndMT conversion process (even though shRNA for Twist showed some effect)?! The same question is about FoxC2, which is indicated as significantly changed in Supplementary Table 3. Do the authors see alterations of FoxC2 expression in HUVECs under 4h shear stress?.

Answer 1: We have published a study that is cited in this manuscript with detailed spatial analysis of all genes related to acute shear stress at 4 hours in vein segments. Based on this and on our spatial data presented in this paper, we aimed to look specifically at TWIST2 and TWIST 1. TWIST 2, as we presented in the paper, was significantly upregulated under acute high shear. We did not observe significant upregulation of SNAIL in our cluster, and we thus did not consider these as EndMT cells. Studying SNAIL is of interest but was not the focus of this study. TWIST is known to regulate TGF-mediated EndMT, however, there are many other triggers to EndMT, and they are beyond the scope of one study. All the genes that are significantly upregulated in this cluster are shown in supp data and all other genes related to EC, SMC and Fibroblasts are already published in our previous paper (10.3390/ijms251910368). We did not look at FOXC2 as there was no need for that in this study, but it will be interesting to look at this in different work; however, our net work analysis reported that it is related to TWIST 2, the same as FOXC1, REL and others which can be looked at in the future or by other groups but not the scope of this study.

Comment 2: General comment and question. For me personally it is a bit difficult to accept the authors data that 4h of a stimulus (shear stress) is enough even for the new protein expression (Vim IFL data on Fig.1). To convert HUVEC into EndMT-like cells I used minimum 24h of a treatment (silencing of some gene - data unpublished). Therefore, my question is: How stable are those EndMT-related changes in cultured HUVECs, observed after 4h of shear stress? If to remove shear stress after 4h and change to static conditions do the cells revert the expression of those markers (Twist, Vim, VEC, Alk5, pSMAD RNA and protein) to the "baseline" level?. When the authors observe overexpression or suppression of a few genes (Vim, Twist, VEcad) does it mean that the real EndMT occurs? Can it be just a few EndMT-related genes (and maybe thousand of unreleated genes, which authors did not analyzed)? HUVECs undergone EndMT might change their morphology (from cobblestone to spindle-shape) and motility. Did the authors observed these effects? The more comprehensive assessment of the cells after the treatment can provide better characterization of the system.

Answer 2: thank you. The data clearly support changes in protein expression, and this is not a unique study to demonstrate changes acutely. We have cited different studies by our group and others that showed that an acute stimulus can lead to early protein expression. It is not known how stable is these changes and not the question we set to answer, it is reasonable to accept that if the stimulus continue then the process of EndMT will continue and cells changes will become more stable as been demonstrated by others who looked at EndMT in vein grafts and showed and essential role for it which we discussed in our discussion. Please note that different stimuli can impact cells differently and it will be interesting to compare in the future the effect of various stimuli on response, but this is not the scope here. With regards to morphological changes in cells, this can be done in HUVEC, but it is not clear how this can be accomplished in sections of tissue, taking into account variation in sections, etc..

Our work aimed to look at the impact of dex pretreatment on EndMT in vein grafts which is already recognised and well known event that occur in veins after CABG.

Please note also our data in supp data when we noted that the brief single treatment was sufficient in suppressing EndMT at 10 days post-culture (supplementary Figure S4 D). Interestingly, when we compared the role of low flow on the activation of LSV ECs, we noted that low flow did not trigger any EndMT changes in LSV ex-vivo (supplementary Figure S4 E). Thus, we conclude that a brief pretreatment with Dex (60 min) can suppress EndMT activation in LSV ECs in response to shear stress by suppressing the TGFβ /SMAD1 pathway.

Comment 3: Fig.S1B,E: Silencing of Twist1 and Twist2 is only 50%! It is a bit difficult to believe that so small suppression level of 50% is enough to fully revert the effect of shear stress on Vim and VEcad. And these siRNA for Twist1 and Twist2 are used separately! How is it possible? If you inhibit just 50% of, for example, Twist1 (Fig.S1B), then full Twist2 is still available and also the remaining 50% of Twist1. And vice versa, when you inhibit Twist2 (only 50% --> Fig.S1B). Did the authors try to inhibit simultaneously both Twist 1 and 2? 

Answer 2: This is a very interesting question. We added data to show that silencing TWIST1 or TWIST2 will not affect the expression of the others, which suggests to us that there is potential for cross-talk between both of them and that both are required at specific levels in this setting for the function to occur which is not a new concept considering that the regulation some processes requires that Twist1 and Twist2 function as molecular switches to activate and repress target genes by employing several direct and indirect mechanisms. (10.1093/nar/gkq890) with evidence of both being required to coordinate responses (10.4049/jimmunol.1402808)

We have added a section in the paper about this. We don’t think based on this co-silencing will add any new information.

Comment 3: General question for the cell experiments and for saphenous vein:
Do the authors link proinflammatory changes and EndMT? One induces the other or they are parallel processes? Proinflammatory changes might be more relevant because they directly induce attraction conditions for the inflammatory cells, which leads to pathological consequences in the wall of a vein (or artery)? I think it needs to be carefully discussed, as I, for example, am not so sure if the EndMT-related changes are not just a secondary ones, while the proinflammatory changes might be the utmost important. Did the authors try to analyze cytokine profiles in HUVECs and LSV tissues after shear stress?

Answer 3: This study is looking at TWIST-related EndMT, the role of Dex pretreatment in suppressing EndMT. The cross-talk between inflammation/ other EndMT pathways is not the scope of this study, but is of interest to look at in the future. Inflammation may play a key role, but this does not mean that EndMT is not occurring and can be triggered by many pathways. We have analysed the whole gene profile in vein segments exposed to arterial haemodynamics in our previous paper, which is cited above and in the manuscript, and we conducted detailed pathways analysis and gene connections in both papers. Of interest, the pathway analysis clearly showed a significant importance for the regulation of biological, cellular and developmental processes.

Round 2

Reviewer 3 Report

Comments and Suggestions for Authors

The authors sufficiently addressed all my concerns. I have no further questions.